# Piezoelectric Impact Energy Harvester Based on the Composite Spherical Particle Chain for Self-Powered Sensors

**DOI:** 10.3390/s21093151

**Published:** 2021-05-01

**Authors:** Shuo Yang, Bin Wu, Xiucheng Liu, Mingzhi Li, Heying Wang, Cunfu He

**Affiliations:** Faculty of Materials and Manufacturing, Beijing University of Technology, Beijing 100124, China; ys@emails.bjut.edu.cn (S.Y.); wb@bjut.edu.cn (B.W.); lmz1001@emails.bjut.edu.cn (M.L.); whyme2015@emails.bjut.edu.cn (H.W.); hecunfu@bjut.edu.cn (C.H.)

**Keywords:** energy harvester, composite granular chain, solitary waves, energy trapping, self-powered system

## Abstract

In this study, a novel piezoelectric energy harvester (PEH) based on the array composite spherical particle chain was constructed and explored in detail through simulation and experimental verification. The power test of the PEH based on array composite particle chains in the self-powered system was realized. Firstly, the model of PEH based on the composite spherical particle chain was constructed to theoretically realize the collection, transformation, and storage of impact energy, and the advantages of a composite particle chain in the field of piezoelectric energy harvesting were verified. Secondly, an experimental system was established to test the performance of the PEH, including the stability of the system under a continuous impact load, the power adjustment under different resistances, and the influence of the number of particle chains on the energy harvesting efficiency. Finally, a self-powered supply system was established with the PEH composed of three composite particle chains to realize the power supply of the microelectronic components. This paper presents a method of collecting impact energy based on particle chain structure, and lays an experimental foundation for the application of a composite particle chain in the field of piezoelectric energy harvesting.

## 1. Introduction

An intelligent monitoring system consisting of environment, mechanical vibration, and structural health conditions generally employs many low-power sensors. In some monitoring scenarios, external power supply is not available for the sensor network, so the intelligent monitoring system needs to be self-powered. PEHs are important optional devices for converting mechanical vibration energy into electrical energy. Numerous types of cantilever- and nonlinear-based PEHs have been applied in the construction of self-powered systems. The performance of the PEHs can be improved through designing novel mechanical structures. For instance, the resonance frequency of the cantilever-based PEHs can be tuned to be close to the external vibration frequency so that the energy harvesting efficiency can be improved [1,2,3,4]. The unique nonlinear vibration characteristics can stimulate nonlinear-based PEHs to produce larger deformation [5,6,7,8]. Most of the reported PEHs so far focus on harvesting the energy of alternating vibration, but the collection of instantaneous energy generated in the process of collision or impact has seldom been reported. In order to endow the PEHs with the ability to harvest energy generated by collision or impact distributed in the scene of rich impact load, such as a bridge or railway, it is necessary to develop new mechanical structures.

The one-dimensional chain of spherical particles is a special carrier of stress waves. An impact imposed onto a vertical chain of homogeneous spherical particles can generate strong nonlinear solitary waves propagating along the chain. Based on the balance of strong nonlinearity and dispersion, the monopolar solitary waves are stable [9]. During the propagation process along a chain composed of metal particles, the energy of solitary waves experiences low attenuation. The nature of low attenuation of solitary waves makes the one-dimensional chain of spherical particles an ideal option for collision or impact energy transmission. By replacing several spherical particles of the chain with energy-harvesting particles, a PEH of collision or impact energy can be developed. Li et al. [10,11,12] constructed an impact PEH with homogeneous particle chains. In their design, the solitary waves propagated along the chains emitted into an elastic base material, and a self-focusing of solitary waves in the base material was realized through optimizing the chains. PEHs were attached near the focal area at the surface of the base material and the high energy harvesting efficiency was expected. The design had two drawbacks. First, most of the energy of the solitary waves was bounced back to the impact source at the interface between the chain and the base material. The quantity of the energy transmitted to the PEHs for harvesting was, therefore, limited. Furthermore, the energy of point impact imposed at the top surface of the base material propagated together with the spherical elastic waves. Only a small part of the energy of the spherical elastic waves could be focused for harvesting. More importantly, it was difficult for the PEHs to re-harvest the energy of reflected solitary waves and the multiple-reflected elastic waves in the base material.

To overcome the drawbacks of the PEHs constructed with a one-dimensional chain of spherical particles reported in previous studies [13,14,15], a novel design of PEHs based on the composite spherical particle chain was introduced and experimentally verified in this study. First, the PEHs were directly embedded into the chain so that the energy of solitary waves transmitted through PEHs could be significantly increased as compared with the previous results [16,17,18]. Second, a three-section chain of heavy-light-heavy was used to form an energy oscillation cavity, which was defined by the interfaces between the light middle chain and two heavy chains. As illustrated in previous studies [19,20], the solitary waves trapped in the energy oscillation cavity were reflected back and forth many times. If the PEHs were placed in the light chain of the energy oscillation cavity, they realized multiple energy collections to improve their energy harvesting efficiency. The better performance of composite spherical particle chains compared with homogeneous particle chains in constructing PEHs was also reported in a previous study [21].

The energy harvesting characteristics of composite particle chains need to be further explored. Firstly, the environmental vibration is usually surface excitation rather than point excitation, so the energy harvesting efficiency can be further improved by increasing the energy harvesting area. Secondly, after optimizing the excitation parameters and dimension parameters, the normalized charging speed is not significantly improved. Thirdly, in previous studies, only the voltage changes at both ends of the energy storage capacitor were tested, but the power supply requirement of the micro-sensor was not tested.

In the study, in order to improve the charging speed and energy harvesting density of PEHs, we studied the structure of composite spherical particle chain, energy harvesting performance, and the power supply performance of the energy harvester to the wireless sensor. Firstly, we established a simulation model of the PEHs of a composite spherical particle chain. We then theoretically and experimentally confirmed the energy trapping phenomenon. Secondly, PEHs based on the array particle chains were designed and built to explore their performance. Finally, a self-powered sensor system composed of PEHs and a wireless microelectronic sensor was designed and built to charge the energy storage capacitor and test sensor parameters. The study provides the experimental basis for the engineering application of PEHs.

## 2. Simulation Model of PEH

The model of a PEH based on a composite spherical particle chain is shown in Figure 1a. The composite spherical particle chain was composed of three sections and one energy harvesting particle arranged in the middle section. All the particles were of the identical diameter D and the number of particles in each section was N. The material of the particles in the two chains at the head and tail was the same, and had higher density (ρ) and Young’s modulus (E) than the particles in the middle section. The particles of the head and tail sections are referred to as heavy particles, and the particles of the middle section are called light particles.

One piezoelectric wafer was glued with two hemispherical particles to form a sandwiched energy harvesting particle. When the chain was impacted by a freely falling heavy particle, which was used to simulate an external collision or impact energy source, solitary waves were generated in the first section of the heavy particle chain and then transmitted into the light particle chain. Most of the solitary waves’ energies were reflected back at the interface between the light and heavy chains (referred to as light-heavy interface). The U-turn solitary waves encountered another light-heavy interface and were reflected back again. The solitary waves reflected back and forth in the light chain seemed to be trapped by two light-heavy interfaces. Therefore, the light chain in the proposed composite spherical particle chain could be treated as a solitary wave’s energy oscillation cavity. Such unique phenomena were reported in previous studies [16]. The energy oscillation cavity is of great benefit to the PEH because the PEH is capable of harvesting the energy of solitary waves multiple times.

The discrete dynamic model proposed in previous studies [22,23,24] was used to predict the propagation behavior of solitary waves in the composite chain. The contact force imposed on the energy harvesting particle of the PEH was transformed into the output voltage through a G-type piezoelectric equation. A capacitor charging circuit simulated the energy harvesting process. The energy harvesting particle acted as an output voltage source (the energy harvesting circuit run in the Simulink of MATLAB software is shown in Figure 1b). The piezoelectric element was equivalent to capacitor C0 and resistance R0 connected in parallel. The voltage source V was connected in series with the resistance R0. Two diodes of D1 and D2 formed a half-bridge rectifier circuit. The voltage signal after rectification was fed into a capacitor of C1 once the switch K was closed for restoring the electric energy. A voltmeter and an oscilloscope were used to measure and display the voltage of the energy storage capacitor, respectively.

In this study, heavy and light particles were respectively made of steel (E = 200 GPa, υ = 0.30, ρ = 7860 kg/m^3^) and aluminum (E = 69 GPa, υ = 0.27, ρ = 2700 kg/m^3^) and had a diameter D of 10 mm. The bottom of the composite chain mechanically contacted with an aluminum block with a dimension of 100 × 100 × 10 mm. The number of particles in each section was the same (N = 10). The diameter and thickness of the piezoelectric disk were 10 mm and 1 mm, respectively. The piezoelectric strain constant was d = 30 × 10^-3^ Vm/N.

The case of the homogeneous chain of steel particles was also investigated for the purpose of comparison. A freely falling steel particle impacted the chain at a speed V_0_ of 0.2 m/s. The predicted and measured output voltage signals of energy harvesting particles in the homogeneous chain of steel particles and the composite chain are shown in Figure 2a,b.

From the Simulink module, the parameters of the electronic components are listed in Table 1. When the switch K was in the off state, the open circuit voltage waveform after rectification was as shown in Figure 3a. When the switch K was closed, the voltage of the energy storage capacitor could be measured. A total of 1000 impacts were investigated for predicting the rising trend in the voltage of the energy storage capacitor. Figure 3b demonstrates the voltage of the energy storage capacitor in the entire charging process.

In Figure 2, the energy trapping phenomenon can be observed when a solitary wave propagates in the composite particle chain. It can be seen from Figure 3b that the rectifier circuit can harvest the energy carried by each tiny signal in Figure 3a, so the tiny energy in the energy trapping process can be harvested and collected.

## 3. Experimental Verification

### 3.1. Experimental Set-Up

The vibration form of the external environment is usually surface vibration rather than point vibration. Therefore, in order to increase the energy harvesting area and improve the energy harvesting efficiency, the number of particle chains can be increased. Firstly, the experimental set-up of the PEH based on array composite particle chains was designed and built (Figure 4a). The sleeve was a five-hole structure, with an outer diameter of 50 mm and an inner diameter of 10.1 mm. A groove with a cross-sectional area of 2×2 mm was processed in each hole as the leading groove of the energy harvesting particle lead, (Figure 4(aI)). The parameters of spherical particles (material, diameter, and number) and the diameter of piezoelectric patches are consistent with the data in Section 2. The electromagnetic vibration exciter (DY-345, Moluoge, Ruzhou, China) was placed on the top bracket of the particle chains. In order to simulate the external surface vibration and ensure the synchronal excitation of the array particle chains, a circular steel sheet with a size of 50×2 mm was fixed on the end of its output shaft as the excitation device (Figure 4(aII)). The function generator (AFG3021B, Tektronix, Beverton, OR, US) output sine wave signals with a fixed frequency and amplitude to the power amplifier (TPA3116, Clover Audio, Shijiazhuang, China), which provided a current to drive the electromagnetic exciter. The energy-harvesting particles were composed of two metal cylinders with a diameter of 10 mm and a thickness of 5 mm and a PZT5H circular piezoelectric patch. The energy harvesting particles were located in the center of the particle chains and the open circuit voltage signal could be observed and recorded by an oscilloscope (MDO4054c, Tektronix, Beverton, OR, US).

The number of each hole of the five-hole array particle chain sleeve is shown in Figure 4b. Figure 4b shows the positions of particle chains with different numbers of particle chains.

### 3.2. Performance Test

The single signal generated by the PEH is characterized by unstable voltage and low current, so the harvested energy under a single excitation is not enough to supply power to low-power microelectronic components. In order to meet the power requirements of microelectronic sensors, the energy harvesting device needs to store the electric energy generated by continuous excitation in the energy storage element, and then discharge it directly via the energy storage element. Therefore, it was necessary to ensure the stable energy harvesting characteristic of the particle chain in the process of continuous impact. The excitation frequency was fixed at 10 Hz and the maximum peak open circuit voltage of the output signal was adjusted to 10 V. The changes in the energy harvesting voltage signals under 120 s of continuous excitation were recorded (Figure 5a). In order to ensure the consistency of open circuit voltage waveform in different time periods, partial enlarged signals at three different time periods were extracted (Figure 5b).

In order to observe the fluctuation range of the signals more clearly, the peak voltages of the incident wave, the reflected wave and the two instances of energy trapping in the partial enlarged diagram were extracted (Figure 6a). Based on the sorted data in Figure 6a, the average values, standard deviations, and error bars of maximum value are shown in Figure 6b.

In the stability test of the system under continuous excitation, the peak voltage of open circuit voltage showed good stability (Figure 5a). In partial amplified signals, the signals generated by each excitation were not superimposed on each other and the waveforms in different time periods showed high consistency (Figure 5b). The peak values of typical waveforms were basically stable (Figure 6). The peak value of the incident wave was the most stable, indicating that the energy of the input particle chain had good consistency. Therefore, the energy capture result was reliable under continuous excitation.

The PEH based on a particle chains array was composed of five identical composite particle chains placed in five holes of the cylindrical sleeve. The five holes were independent from each other, so the energy harvesting effect of each particle chain was the same in principle. Firstly, the energy-harvesting characteristics of a single particle chain in the central hole were explored. Secondly, the peak voltages of the open circuit voltage signals were fixed at 10 V. Then, different resistors were connected at the output end of the PEH and the peak voltage change at the ends of the resistor was measured and recorded by an oscilloscope. According to the peak voltage and external resistance, the peak current variation with the external resistance could be calculated (Figure 7a). Finally, according to the peak voltage and peak current, the power adjustment with different external resistances was calculated (Figure 7b).

The peak voltage decreased with the increase in external resistance, whereas the peak current showed the opposite trend (Figure 7a). With the increase in external resistance, the maximum output power firstly increased and then decreased (Figure 7b). When the external resistance was about 30 kΩ, the maximum power reached 1.05 mW.

Figure 8a–c show the open circuit voltage signals generated by the PEH under the three arrangement modes of particle chains under a fixed peak value of the open circuit voltage, which was 10 V. Energy storage capacitors were charged separately and the voltage at both ends of the capacitor was measured every 10 min. Figure 8d shows the trend in the voltage at the ends of the capacitor within 180 min.

When the three kinds of particle chain arrays shown in Figure 8a–c were excited, due to inevitable system error, the peak voltage of the open circuit voltage signal generated by each particle chain was not the same. Therefore, when more than one particle chain was excited simultaneously, it was only necessary to ensure that the highest peak voltage of all harvesting signals was 10 V. By increasing the number of particle chains, the energy harvesting effect was significantly improved, but the voltage at the ends of the energy storage capacitor did not change linearly with the increase in the number of particle chains (Figure 8d). It was impossible to ensure that the peak voltage of the signal generated by each particle chain simultaneously reached the expected amplitude (10 V), so the growth rate of the charging speed gradually became slower with the increase in the number of particle chains. Regardless of the position of particle chains, when the number of particle chains increased to four or more, only three particle chains could be excited simultaneously. The result might be interpreted as follows. Unless the contact point between the exciting steel sheet and the particle chain was in the same plane, the particle chain could not be excited by the exciting steel sheet. However, limited by processing or assembly accuracy, the height of each particle chain was not uniform. According to the principle that three points determine a plane, when the number of particle chains exceeded three, only three particle chains could be excited simultaneously.

## 4. Application

The micro-sensor self-powered supply system was built to explore the engineering application performance of the PEH. Its schematic diagram and photo are shown in Figure 9a,b, respectively. The power supply process is described as follows. Firstly, the electric signal generated by the PEH in the array composite 3-particle chains was collected in the energy storage capacitor C1 via the energy harvesting circuit. At this time, the switch K between the energy storage capacitor and the micro-sensor was in the off state. When the voltage of the energy storage capacitor was charged to a certain value, the switch K was closed and the energy storage capacitor started to supply power to the low power wireless module composed of the micro-sensor (GY-39, Yu Song Electronic, Shenzhen, China) and Bluetooth module (JDY-31, Yu Song Electronic, Shenzhen, China). Then the data generated by the sensor were transmitted to the host computer through the Bluetooth module and displayed by LabVIEW. The host computer could achieve real-time display of temperature, humidity, light intensity, and atmospheric pressure parameters after signal processing. The signal acquisition diagram of the host computer is shown in Figure 9c.

Firstly, the PEH charged the 6.8 mF energy storage capacitor to 5 V. Then, the switch was closed and the energy storage capacitor discharged to supply power to the wireless sensor module. In the charging and discharging processes, the trend in the voltage of the energy storage capacitor was as shown in Figure 10a. Compared with the charging process, the discharge process was relatively short. In order to observe the trend in the capacitor’s voltage in the discharge process more clearly, the partial enlarged diagram of discharge process is shown in Figure 10b. The self-powered supply system ran twice. The two sets of data received by the host computer are shown in Figure 10b.

In the above results, the PEH met the basic engineering required application performance of energy collection, conversion, and utilization. In order to improve the engineering application performance, the energy storage element was replaced by a super capacitor with a capacity of 100 mF. If the capacitor continued to be charged by the PEH, it was only necessary to increase the charging time. In order to explore the power supply effect of the super capacitor more efficiently, the super capacitor was charged to 5 V by a regulated external power supply. The trend in the voltage of the energy storage capacitor during the discharge process is shown in Figure 11a. The capacitor with a large capacity could continuously supply power to the wireless sensor module 13 times. The data showing this are shown in Figure 11b,c.

## 5. Conclusions

In this study, a PEH based on the array composite spherical particle chain was developed to collect impact energy. First, through the establishment of the simulation model, the mechanism of energy harvesting was explored and then an experimental system was built to test the performance. Finally, the power supply to the sensor was realized.

The unique energy trapping phenomenon in the composite particle chain was observed in the calculation results of the simulation model and the experimental verification results of the PEHs.

The stability of the PEH could be guaranteed under continuous impact excitation. When the power adjustment was measured, the maximum output power firstly increased and then decreased with the increase in the external resistance, and the maximum power reached 1.05 mW when the external resistance was 30 KΩ. The peak voltage decreased with the increase of the external resistance, whereas the peak current showed the opposite trend.

For the PEH based on array multi-particle chains, the energy harvesting effect increased with the increase in the number of particle chains. Limited by the processing and assembly accuracy of the experimental system, the maximum number of particle chains was three. The average energy harvesting effect of each particle chain decreased with the increase in the number of particle chains.

The test results of the self-powered supply system constructed with the PEH, weather sensor, and Bluetooth module indicated that the PEH met the power supply requirements of the system. This study can be used to guide the engineering application of PEHs.

## Figures and Tables

**Figure 1 sensors-21-03151-f001:**
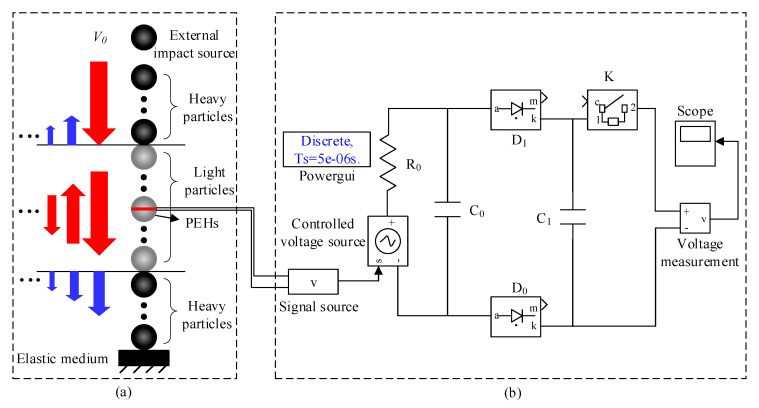
Simulation model of PEH:(**a**) composite particle chain and (**b**) energy trapping circuit based on Simulink.

**Figure 2 sensors-21-03151-f002:**
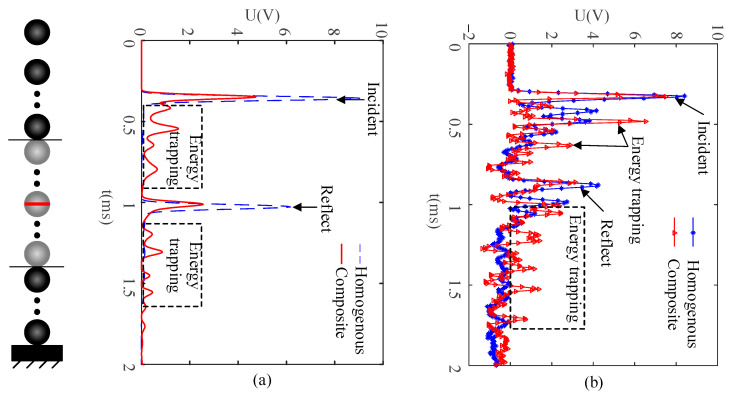
(**a**) Predicted and (**b**) measured output voltage signals of PEH.

**Figure 3 sensors-21-03151-f003:**
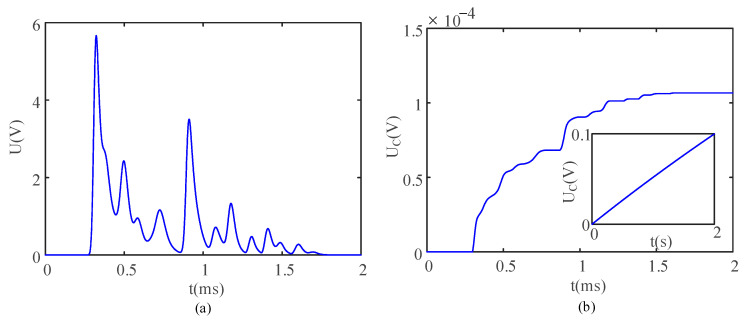
(**a**) Typical voltage waveform after rectification and (**b**) the voltage of the energy storage capacitor in the charging process.

**Figure 4 sensors-21-03151-f004:**
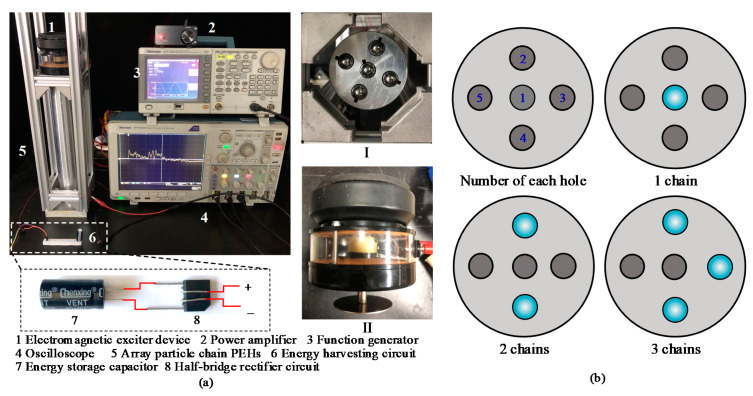
(**a**) Experimental set-up of the PEH based on multiple composite chains of particles and (**b**) layout of the impacted chains.

**Figure 5 sensors-21-03151-f005:**
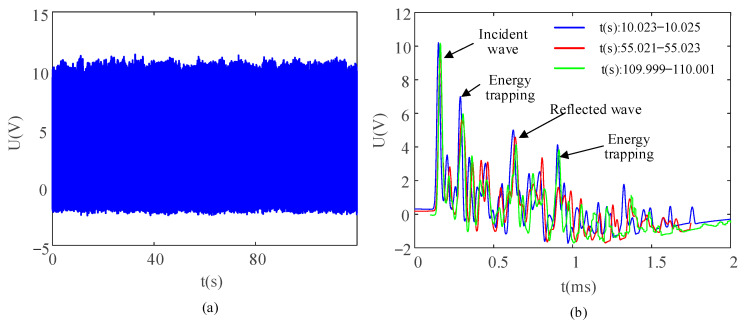
(**a**) Recorded voltage signals of the PEH in the impact process of 120 s and (**b**) typical waveforms of different time periods.

**Figure 6 sensors-21-03151-f006:**
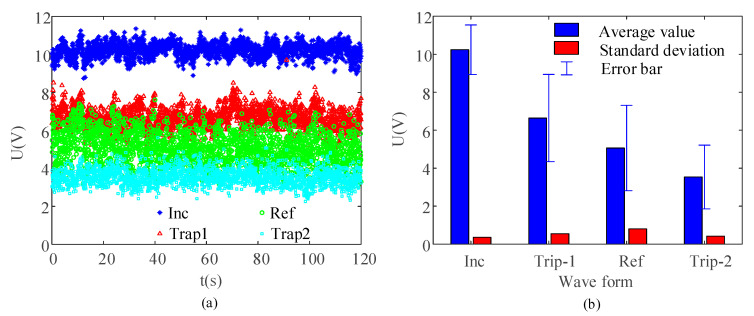
(**a**) Recorded peak-voltage signals of the capacitor in the impact process of 120 s and (**b**) fluctuation data of peak voltage.

**Figure 7 sensors-21-03151-f007:**
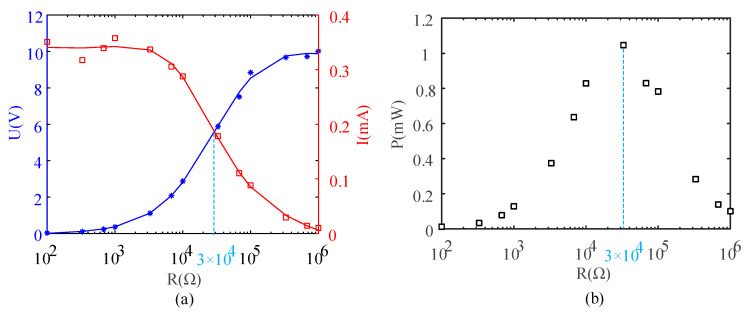
Influences of external resistance on (**a**) voltage and current peaks and (**b**) power.

**Figure 8 sensors-21-03151-f008:**
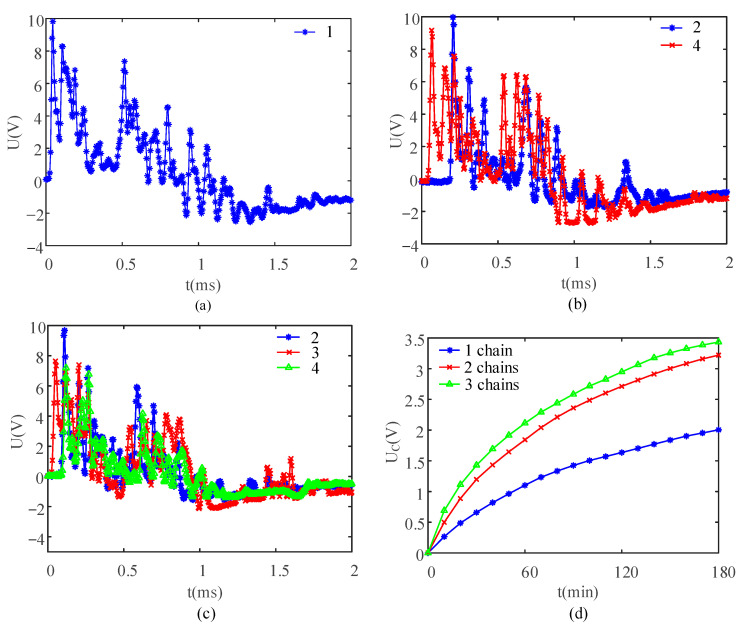
Influences of the number of chains on the capacitor charging process: (**a**–**c**) typical open circuit voltage received by single, two, and three composite chains and (**d**) capacitor charging process in all three cases.

**Figure 9 sensors-21-03151-f009:**
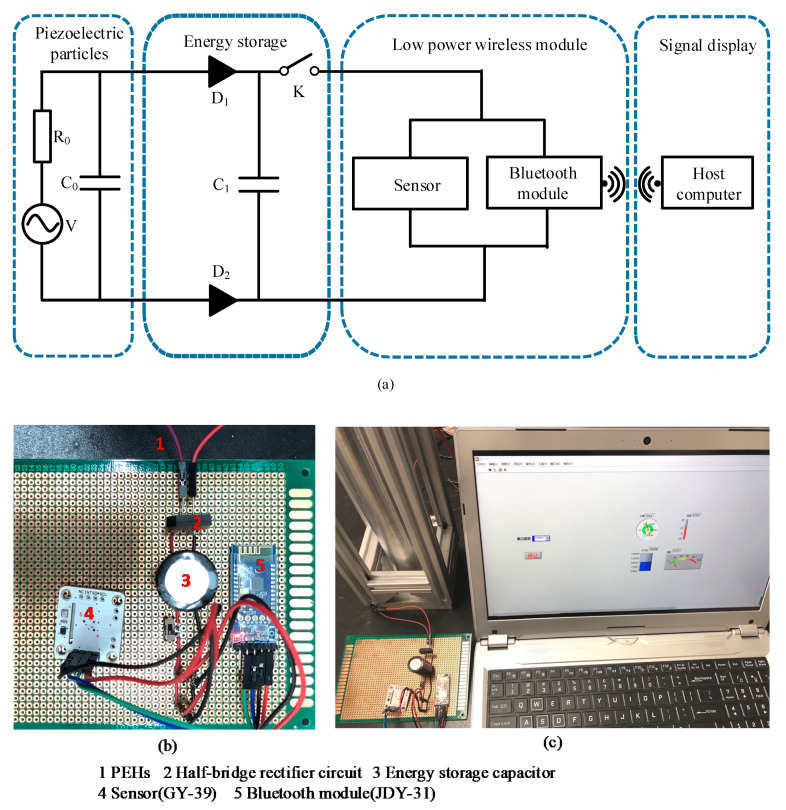
(**a**) Schematic diagram and (**b**) photo of the self-powered wireless meteorological sensor, and (**c**) the signal acquisition diagram of the host computer.

**Figure 10 sensors-21-03151-f010:**
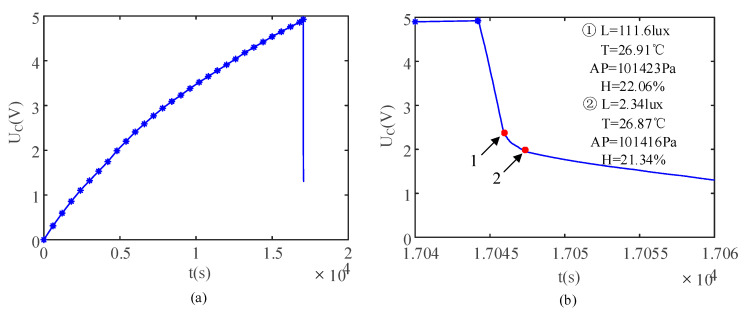
Voltage variations in the energy storage capacitor in (**a**) the charging process and (**b**) discharging process.

**Figure 11 sensors-21-03151-f011:**
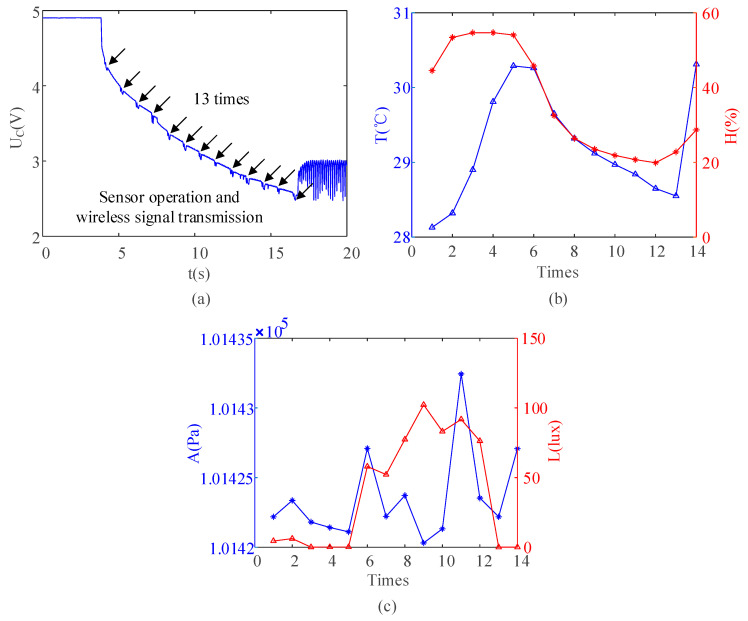
(**a**) Discharging progress with capacitance of 100 mF and (**b**,**c**) the data of the sensor.

**Table 1 sensors-21-03151-t001:** Elements in the energy harvesting circuit.

Elements	R_0_ (Ω)	C_0_ (C)	D_1_ D_2_				C_1_ (F)
			Ron (Ω)	Vf (V)	Rs (Ω)	Cs (F)	
Parameters	1 × 10^4^	51 × 10^−9^	0.01	0.05	5 × 10^12^	2.5 × 10^−7^	1 × 10^−3^

## Data Availability

Not applicable.

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
