# Peer review of "Piezoelectric Impact Energy Harvester Based on the Composite Spherical Particle Chain for Self-Powered Sensors"

_sensors, 2021, doi:10.3390/s21093151_

Round 1
Reviewer 1 Report
Authors proposed a piezoelectric energy harvester (PEH) based on the spherical particle chain supporting strongly nonlinear Nesterenko solitary waves. Numerical simulation of the collection, transformation and storage of impact energy and experimental verification are in a reasonable agreement. The experimental set up was used to test the performance of the PEH, including the stability of the system under continuous impact load, the influence of resistance on the energy harvesting characteristics, and the influence of the number of particle chains on the energy harvesting efficiency.
The article has interesting results and it can be published after significant corrections are introduced. Some suggestions are presented below. My general observation is unprofessional referencing, which is not reflecting state of the art and contribution of other researchers.
- The sentence on the first page “The one-dimensional chain of spherical particles is a special carrier of stress waves. An impact imposed onto a vertical chain of homogeneous spherical particles can generate highly nonlinear solitary waves (HNSWs) propagating along the chain. The HNSWs are unipolar and non-dispersive” can be confusing.
First, these highly nonlinear solitary waves are relatively new phenomenon and appropriate references should introduce readers to the unique nature of these novel solitary waves. The proper approach would be placing reference [25] in this paragraph and may be adding reference to recent review Nesterenko VF. 2018 Waves in strongly nonlinear discrete systems. Phil. Trans. R. Soc. A 376: 20170130. http://dx.doi.org/10.1098/rsta.2017.0130.
Secondly, these strongly nonlinear solitary waves are resulting from a balance of strong nonlinearity and strong dispersion. Thus, it is technically incorrect to call them “non-dispersive”.
Finally, researchers who were pioneers in the corresponding area coined a special name for these unique solitary waves, which is used in many papers unlike the awkward acronym “HNSW” used by some authors and in this article. History of research in this area and corresponding naming can be found in review by a pioneering researcher Nesterenko VF. 2018 Waves in strongly nonlinear discrete systems. Phil. Trans. R. Soc. A 376: 20170130.
- The following sentences “During the propagation process along the chain, the HNSWs, therefore, experience low attenuation. The nature of low attenuation of HNSWs makes the one-dimensional chain of spherical particles be an ideal option for collision or impact energy transmission.” again are confusing and in fact misleading, especially in relation to storage of impact energy. Low attenuation of the Nesterenko solitary waves is observed only for chains made of metal, glass, or ceramic beads (e.g., stainless steel beads). In case of the chain made attenuation of these solitary waves is rather strong. Thus, level of attenuation is determined not by intrinsic properties of solitary waves, but rather properties of the beads.
- The references in the sentence “As illustrated in previous studies [18-19], the HNSWs trapped in the energy oscillation cavity were reflected back and forth many times” incorrectly reflects the state of the art. In fact, trapping of waves inside composite chain was reported in reference [16]. Reference [18] does not look like appropriate to the phenomena of wave trapping in composite chains.
- The sentence in the 3rd page “Such unique phenomena have been reported in the previous studies [21-22]”. These references do not reflect state of the art. A similar phenomenon was observed in earlier publications, e.g., in [16].
- The sentence in the 3rd page “The discrete dynamic model proposed in previous studies[23-25] can be used to predict the propagation behavior of HNSWs in the composite chain” is confusing. Reference [23] is not related to wave propagation in the chain, it is related to static contact interaction between spherical elastic particles, which can be adopted to wave propagation under certain conditions explained in [24,25].
Reviewer 2 Report
It is completely incomprehensible why the actual main parameter, the force of the incident wave, is absolutely not characterized.
How big is their amplitude?
How does the system behave when the amplitude of the wave changes? Instead, it is only said that the excitation frequency is fixed at 10Hz and the open circuit voltage is limited to 10V. In the latter parameter, however, the "how" is missing. Nice that energy was trapped, but at what price?
Fundamental deficiency:
The lack of relationships between input and output variables makes all statements about output variables irrelevant. The evaluation of all subsequent statements can therefore only concern their specifics due to the basic deficiency.
The behavior to Fig. 7 described in Section 1 on page 8 has been known for over 100 years - it is called power adjustment.
Figure 8 on the same page is useless because of the basic flaw and is incomprehensible. If 3 similar harvesters are connected in parallel, then their output size must be reduced to 1/3 each. Anything else would be a perpetual motion machine. However, since the input variable is not described, the diagrams are meaningless.
The application example described is understandable.
Fig. 10b is incomprehensible. Why is the voltage drop from the 2nd measuring point (5V) much greater than that between points 1 and 2?
How can one draw such a discharge curve on the basis of 2 values?
What should diagrams 11a-b) indicate in relation to the PEH?
A fundamental revision of the paper is imperative. The main focus here is on the relationship between input variables and output variables.
Reviewer 3 Report
With reference to the manuscript titled “Piezoelectric impact energy harvester based on the composite spherical particle chain for self-powered sensors” (manuscript ID: sensors-11505448), I believe that the manuscript is well written, providing good contents in terms of quality and originality, and supporting their conclusions with the experimental results. However, some content lacks and formal issues are found in the manuscript for improving the legibility of the paper. In particular, the main issues found in the manuscript are the following:
- The authors are suggested to review the abstract to highlight the main contributions and results of the carried out experimental activity.
- The authors are recommended to extend the related works section with recent scientific works.
- The authors are recommended to review the whole paper to comply with the journal template (for further info, see: https://www.mdpi.com/journal/sensors/instructions )
- Relatively to the Introduction, the authors are suggested to highlight the main contributions and results of carried out work, along with possible application scenarios of considered PEHs.
- The authors are advised to add a summary of the manuscript structure to improve the manuscript’s legibility at the end of the introduction.
- The authors are advised to review the whole manuscript to improve the English language and correct typos (e.g. the titles and paragraph numbers at line 152 and 179 are the same).
- Line 189. “PEH was adjusted to 10V” Please specify. Do you mean that the peak voltage at the output of the conditioning section is set to 10V?
- The authors are suggested to specify details (model, manufacturer, specs, etc.) about all used instruments, components, hardware and software tool involved in the experimental activity.
- “Compared with the charging process, the discharge process is relatively short”, What’s the power consumption of the sensor module? Also, you can calculate the autonomy of the device.
Round 2
Reviewer 1 Report
The corrected article is acceptable for publication.
Author Response
According to your suggestions, the revised paper has been greatly improved. We would like to thank you again for all the comments of our manuscript (No.: 1150548) entitled “Piezoelectric impact energy harvester based on the composite spherical particle chain for self-powered sensors”.
Reviewer 2 Report
Please see my comments in the attached file.

Reviewer 3 Report
After a careful review of the manuscript titled "Piezoelectric impact energy harvester based on the composite 2 spherical particle chain for self-powered sensors", I'm quite satisfied with the integrations/modifications carried out on the manuscript, increasing the detail degree about the methodology employed to characterize the developed energy harvester and, in general, the legibility of the manuscript.
For these reasons, I suggest the acceptance of the manuscript in the present form
Author Response

(The authors gave the same response as above.)

Round 3
Reviewer 2 Report
As long as you don't say nothing about the input power, you can not count with a positive response. The paper is owful in this shape